# Genome-Wide Identification and Expression Analysis of the Phosphate Transporter Gene Family in *Zea mays* Under Phosphorus Stress

**DOI:** 10.3390/ijms26041445

**Published:** 2025-02-09

**Authors:** Yueli Wang, Ronglan Li, Yuanhao Guo, Yan Du, Zhiheng Luo, Yuhang Guo, Tobias Würschum, Wenxin Liu

**Affiliations:** 1State Key Laboratory of Maize Bio-Breeding, National Maize Improvement Center, College of Agronomy and Biotechnology, China Agricultural University, Beijing 100193, China; 2Sanya Institute of China Agricultural University, China Agricultural University, Sanya 572025, China; 3Institute of Plant Breeding, Seed Science and Population Genetics, University of Hohenheim, 70599 Stuttgart, Germany

**Keywords:** low-phosphorus stress, maize, PHT family, stress responses

## Abstract

Phosphorus is one of the key limiting factors for maize growth and productivity, and low-phosphorus stress severely restricts crop yield and stability. Enhancing the ability of maize to grow under low-phosphorus stress and improving phosphorus use efficiency (PUE) are crucial for achieving high and stable yields. Phosphate transporter (PHT) family proteins play a crucial role in the absorption, transport, and utilization of phosphorus in plants. In this study, we systematically identified the PHT gene family in maize, followed by the phylogenetic, gene structure, and expression profiles. The results show that these genes are widely distributed across the 10 chromosomes of maize, forming multiple subfamilies, with the PHT1 subfamily having the largest number. Cis-regulatory element analysis revealed that these genes might play key roles in plant stress responses and hormone regulation. Transcriptome analysis under phosphorus-deficient and normal conditions demonstrated developmental stage- and tissue-specific expression patterns, identifying candidate genes, such as *ZmPHT1-3*, *ZmPHT1-4*, *ZmPHT1-10*, and *ZmPHO1-H3*, involved in phosphorus stress response. This study presents a comprehensive and systematic analysis of the PHT gene family in maize, providing key molecular resources for improving phosphorus use efficiency and breeding phosphorus-efficient maize varieties.

## 1. Introduction

Phosphorus (P) is an essential macronutrient for plant growth and development. It is a component of compounds like ATP and nucleic acids, which transfer energy within cells and support photosynthesis and respiration in plants. Additionally, phosphorus is a fundamental component of DNA and RNA, involved in the transmission and expression of genetic information. Phospholipids, as a key component of cell membranes, maintain membrane integrity and function [1]. Phosphorylation is a crucial mechanism of regulating protein function and influences numerous physiological processes and stress responses in plants [2,3]. Phosphorus also plays a significant role in plant responses to abiotic stressors such as drought, salinity, and high temperatures. Studies have shown that phosphorus deficiency affects stomatal conductance in faba bean (*Vicia faba* L.), thereby influencing plant water use efficiency [4,5]. Moreover, research has demonstrated that phosphorus is important for root development, where plants increase lateral root and root hair growth under low-phosphorus stress to enhance phosphorus absorption [6]. Phosphorus plays a critical role in many aspects of plant growth and development. Phosphorus in the soil mainly exists in the form of inorganic phosphates (e.g., H₂PO₄^−^ and HPO₄^2−^), and its concentration in the soil is typically low, ranging from 0.1 to 10 μM [7]. The concentration of directly usable phosphate forms (soluble phosphates) available to plants is even lower, far below the amount required for optimal growth [8]. Therefore, plants must rely on efficient phosphate uptake mechanisms to meet their growth demands [9,10,11,12,13,14,15].

The PHT family in plants facilitates the absorption and transport of phosphorus. PHT family proteins are responsible for transporting phosphorus from the soil into the plant and redistributing it internally to meet physiological requirements. For example, the PHT1 family has been most extensively studied, with research showing that it is primarily expressed in roots and involved in high-affinity phosphate absorption at the plasma membrane of root epidermal and cortical cells [16,17]. PHT1 transporters are the primary mediators of phosphate uptake, and the number of PHT1 proteins in the plasma membrane increases in response to low phosphate availability [18]. Study has shown that *PvPht1;4* is highly expressed in the leaves and roots of Pteris vittata, with its root transcripts being induced under phosphate deficiency [19]. Additionally, in rice, the expression of the *OsPHT1;1* gene is regulated by the transcription factors *OsWRKY21* and *OsWRKY108*, which bind to the W-box region of its promoter. Overexpression of *OsPHT1;1* enhances the expression of *OsWRKY21* and *OsWRKY108*, resulting in inorganic P (Pi) accumulation [20]. The PHT2 family, in Arabidopsis, participates in phosphate redistribution within chloroplasts, facilitating phosphorus transport and reuse within the plant [21]. The soybean *GmPHT2* gene enhances phosphorus acquisition efficiency, especially under low-phosphate and low-temperature stress [22]. The PHT3 family primarily functions in mitochondria, where it plays a role in phosphate transport essential for energy metabolism [13,23]. Members of the PHT4 family are located in the Golgi apparatus and other organelle membranes, regulating phosphate transport and distribution within organelles [14]. In Arabidopsis, *PHT4;2* acts as an important Pi transporter in root plastids. Its loss results in phosphate accumulation within root plastids, affecting starch synthesis and overall plant metabolism and growth. *PHT4;6*, a phosphate exporter in the Golgi apparatus, is essential for plant development, as its dysfunction causes impaired growth, reduced cytoplasmic phosphate levels, and altered Golgi-related functions and pathogen defense mechanisms [24,25]. The PHT5 family, also known as VPT1, serves as a phosphate transporter in vacuoles [15]. Additionally, in Arabidopsis, the PHO1 family is primarily localized in the endomembrane system of pericycle cells (mainly the Golgi apparatus). The expression patterns of PHO1 homologs suggest that, besides transporting phosphate to vascular tissues, PHO1 may also be involved in phosphate uptake at the cellular level, such as in pollen or root epidermal/cortical cells. Under phosphate deficiency, PHO1 participates in long-distance signal transduction from root to shoot. Research has shown that the *OsPHO1;3* gene in rice is highly expressed in companion cells of the leaf phloem but not in xylem, and its expression is induced under phosphate starvation stress [26,27,28,29,30,31]. The diverse roles of the PHT family in phosphorus uptake, transport, and redistribution underscore their essential contribution to enhancing plant phosphorus efficiency, particularly under phosphorus-limited conditions, by coordinating absorption, internal allocation, and adaptive responses to nutrient stress.

Maize is a globally important food and economic crop, with phosphorus deficiency often limiting plant growth and yield [7,32]. The foliar application of phosphorus fertilizer has been shown to enhance maize drought resistance and promote overall growth and development [33]. However, the global population is projected to reach 10 billion by 2050, creating an urgent demand for increased food production [34]. Without improving phosphorus use efficiency in plants or optimizing fertilizer application methods, an additional 500 million hectares of arable land will be required to sustain this population growth [35,36]. Therefore, utilizing or even enhancing the function of the PHT family to improve the growth capacity and phosphorus utilization efficiency of maize in low-phosphorus environments has become an urgent issue for scientists and breeders.

To this end, a detailed analysis and characterization of the PHT gene family in maize was performed in this study. Although the PHT1 gene family in maize has been studied [37], research on the functions and regulatory mechanisms of the PHT2, PHT3, PHT4, PHT5, and PHO gene families is scarce. In particular, the objectives of this study were (1) to employ bioinformatics to comprehensively analyze the PHT gene family in maize, (2) to characterize these genes and their protein products in order to provide targets for the breeding of maize varieties capable of efficient growth in low-phosphorus environments, and (3) to offer resources for functional gene studies and molecular breeding, promoting agricultural production with enhanced resource utilization and sustainability.

## 2. Results

### 2.1. Identification and Phylogenetic Classification of the ZmPHT Gene Family

In this study, we successfully identified 34 members of the PHT gene family in the maize genome through BLAST-based sequence comparison between Arabidopsis and maize protein sequences. These include 11 members of the PHT1 subfamily, 1 member of the PHT2 subfamily, 5 members of the PHT3 subfamily, 9 members of the PHT4 subfamily, and 4 members each of the PHT5 and PHO1 subfamilies (Appendix A). Among the identified 34 PHT genes, the genes of the PHT1 subfamily were mainly located on chromosomes 1, 2, and 5, while other subfamilies exhibited a more scattered distribution across the maize genome (Figure 1). The amino acid sequence lengths of the 34 PHT genes ranged from 318 aa (*ZmPHT3-2*) to 837 aa (*ZmPHO1-H1*). Within the PHT3 and PHT5 subfamilies, the length ranges were relatively small, from 318 aa to 382 aa and from 692 aa to 699 aa, respectively. The nucleotide sequence lengths of the 34 PHT genes ranged from 1593 bp (*ZmPHT1-7*) to 8998 bp (*ZmPHO1-H4*). The molecular weights (MWs) of the proteins ranged from 34.83 kDa (*ZmPHT3-2*) to 94.28 kDa (*ZmPHO1-H1*). Among them, the PHT3 subfamily exhibited relatively lower molecular weights, while the PHO1 subfamily showed larger molecular weights. The isoelectric points (pIs) ranged from 5.87 (*ZmPHT5-1*) to 9.93 (*ZmPHT4-1*). The pI range of the PHT5 subfamily, from 5.87 to 8.36, spans acidic to basic values, indicating its adaptability to diverse cellular environments and functional versatility [38]. Acidic members likely operate in acidic organelles, while basic members function in alkaline environments, underscoring their ecological and physiological importance in plant phosphorus transport and regulation.

The instability index of the PHT gene family ranged from 25.07 (*ZmPHT3-1*) to 54.44 (*ZmPHT4-2*), suggesting functional diversity within the PHT family, as these genes may be involved in different biological functions. They may play roles in permanent physiological processes, such as maintaining intracellular phosphate levels, or participate in more dynamic or transient phosphate regulation within the cell [39,40]. There are differences in the number of transmembrane helices among the ZmPHT subfamilies (Appendix A). Transmembrane domain predictions were conducted using the TMHMM-2.0 tool. Most genes in the ZmPHT1 subfamily possess 11 to 12 transmembrane helices, indicating that they may play a critical role in transmembrane phosphate transport. The *ZmPHT2;1* gene in the ZmPHT2 subfamily contains 12 transmembrane helices. In contrast, no transmembrane helices were predicted in the genes of the ZmPHT3 subfamily. The genes in the ZmPHT4 subfamily exhibit diversity in the number of transmembrane helices, ranging from 9 to 12, suggesting functional complexity within this subfamily. Genes in the ZmPHT5 subfamily contain 9 to 10 transmembrane helices. Finally, genes in the ZmPHO1 subfamily have fewer transmembrane helices, with only five to six, which may indicate that their role in transmembrane phosphate transport differs from that of other subfamilies.

### 2.2. Phylogenetic Analysis of PHT Proteins

To study the phylogenetic relationships of the PHT gene family across different species, we compared the amino acid sequences of maize PHT proteins with those of the monocot model crop rice, the dicot model plant Arabidopsis, as well as tomato, white lupin, and sorghum, which is closely related to maize (Figure 2). The PHT gene family is divided into six subfamilies, PHT1, PHT2, PHT3, PHT4, PHT5, and PHO1, with the total number of genes ranging from 23 in tomato to 35 in white lupin. Among these six species, the number of PHT gene family members and the distribution across subfamilies vary (Table 1). Except for white lupin and Arabidopsis, PHT1 is the largest subfamily, whereas the PHT2 subfamily has only one member in all species, except in white lupin, which has three members. The phylogenetic tree shows that maize is closely related to the monocot crops sorghum and rice, further supporting the evolutionary relationship among these species.

The PHT1, PHT2, PHT5, and PHO1 subfamilies are mostly localized to the plasma membrane (Figure 2). Other subfamilies are localized to organelles such as the chloroplast (Chl), mitochondria (Mt), and the endoplasmic reticulum (ER). The aliphatic index of most PHT proteins is above 90, indicating that they contain a high proportion of hydrophobic amino acids (such as alanine, valine, leucine, and isoleucine), making them more likely to stably embed in cell membranes. These aliphatic amino acids help stabilize the structure of membrane proteins, ensuring that they maintain stability at the membrane interface [41]. Moreover, the grand averages of hydropathicity values of PHT family proteins are mostly close to zero or positive, indicating that while they are predominantly hydrophobic, they also interact with the aqueous environment inside the cell membrane, facilitating their embedding and functional performance (Appendix A). These characteristics of the PHT family proteins play a key role in transmembrane transport and signal transduction. Their properties not only ensure their stability within cell membranes but also affect their activity and efficiency in the transport of substances across the cell membrane.

### 2.3. Structural Analysis of ZmPHT Genes

Structural analysis of the maize PHT gene family revealed its potential functions and regulatory mechanisms, providing a theoretical basis for genetic improvement and new variety breeding in maize. In-depth analysis revealed that members of the PHT gene subfamilies are highly conserved, but there are significant differences in gene structure among different subfamilies (Figure 3). This finding is partially consistent with the results of Aslam et al. regarding the PHT family members in white lupin [42]. Specifically, the conserved domain of the ZmPHT1 subfamily is Sugar (and other) transporter, the ZmPHT2 subfamily has the PHT family, and the ZmPHT3 subfamily includes two or three mitochondrial carrier protein domains. Additionally, both the PHT4 and PHT5 subfamilies contain the Major Facilitator Superfamily, but the ZmPHT5 subfamily has an additional SYG1/PHO81/XPR1 domain at the N-terminus. Finally, the ZmPHO1 subfamily contains two conserved domains, SYG1/PHO81/XPR1 and ERD1/mammalian XPR1/SYG1.

The number of exons and introns within the ZmPHT genes indicates diversity among the ZmPHT subfamilies (Figure 3). For example, most members of the ZmPHT1 subfamily have one exon, but the *ZmPHT1-3*, *ZmPHT1-7*, and *ZmPHT1-10* genes have two exons, indicating structural diversity even within the same subfamily. The ZmPHT2 subfamily has three exons. The ZmPHT3 and ZmPHT5 subfamilies exhibit high conservation, with genes within each subfamily having the same number of exons, 6 and 10, respectively. The ZmPHO1 subfamily has a higher number of exons: *ZmPHO-H1* has 10 exons, *ZmPHO1-H2* and *ZmPHO1-H4* each have 14 exons, and *ZmPHO1-H3* has 12 exons.

Motif prediction analysis revealed substantial differences in motif composition among the different subfamilies of the maize PHT gene family, while motif composition within the same subfamily showed high consistency (Figure 3 and Appendix A). The conservation and specificity of these motifs may indicate the uniqueness of each subfamily member’s biological functions and regulatory mechanisms. For example, the PHT1 subfamily contains motifs 1 through 8, while the PHT2 subfamily lacks conserved motifs, suggesting that it may play a more independent or distinct role in biological functions. The PHT3 subfamily contains motifs 10 and 13, while the PHT4 subfamily includes motifs 1, 7, 8, 9, and 15. The PHT5 subfamily contains motifs 10, 11, 13, and 14, and the PHO1 subfamily contains motifs 11, 12, and 14. The differences in motif composition among the subfamilies suggest that they may play different roles in biological pathways and functions. However, some motifs appear in multiple subfamilies, such as motifs 7 and 8 in both the PHT1 and PHT4 subfamilies, motif 10 in both the PHT3 and PHT5 subfamilies, and motif 11 in both the PHO1 and PHT5 subfamilies, indicating that these motifs may regulate certain shared physiological processes. In conclusion, the maize PHT gene family exhibits substantial diversity in structure, exon and intron numbers, and motif composition among the different subfamilies, while maintaining high consistency within each subfamily. This characteristic may reflect the functional specificity and regulatory mechanisms unique to each subfamily, while also revealing shared physiological processes indicated by certain motifs present across multiple subfamilies.

### 2.4. Cis-Regulatory Elements in the Promoters of ZmPHT Genes

To predict the cis-regulatory elements in the promoter regions of ZmPHT family members, aiming to reveal the potential roles of these genes within regulatory networks, we analyzed the 2000 bp upstream promoter sequences of ZmPHT genes (Figure 4). The results show that the promoter regions of these PHT subfamily genes are generally enriched with an abscisic acid-responsive element (ABRE), an anaerobic responsive element (ARE), and G-box elements, which are cis-acting regulatory elements involved in light responsiveness, suggesting that this gene family may play a key role in responses to abscisic acid (ABA), environmental stresses, and light signal transduction.

In particular, the promoter regions of the PHO1 subfamily were generally enriched in hormone- and stress-related motifs, especially in *ZmPHO1-H1*, where a large number of low-temperature response elements (LTRs) and GC motifs, which are enhancer-like elements involved in anoxic specific inducibility, were identified. This suggests that this gene may play a significant role in plant stress signaling pathways, particularly under stress such as phosphate starvation. The promoter regions of the PHT1 subfamily showed considerable potential for hormone regulation, especially *ZmPHT1-4*, *ZmPHT1-5*, *ZmPHT1-6*, and *ZmPHT1-8*, where a large number of ABREs, CGTCA motifs—which are cis-acting regulatory elements involved in MeJA responsiveness—and gibberellin-responsive elements (GARE motifs) were detected. This suggests that these genes may exhibit specific activity in response to ABA, methyl jasmonate (JA), and gibberellin (GA) signals. As these hormone response regulations are closely related to adaptive responses under phosphate stress, these genes may play a key role in the adaptation of plants to low-phosphate environments. ABA promotes phosphate uptake under low-phosphate stress by regulating the expression of PHT proteins [43]. The promoter regions of the ZmPHT2 subfamily genes were enriched in light-responsive and hormone-related motifs, while the promoter regions of the ZmPHT3, ZmPHT4, and ZmPHT5 subfamily genes were enriched with G-box and ABREs related to light and ABA responses. This suggests that these genes may play a role in plant responses to environmental stress, especially in light signaling and hormone regulation under phosphate stress. In particular, the promoter regions of *ZmPHT3-2*, *ZmPHT4-6*, and *ZmPHT5-2* were not only enriched in light and ABA response motifs but also stress-related motifs, indicating that these genes may play a regulatory role under multiple environmental stresses.

### 2.5. Synteny and Collinearity Analysis of ZmPHT Proteins

Synteny analysis can reveal the evolutionary history and functional conservation of gene families, and we therefore investigated the syntenic relationships of maize PHT genes to explore their functions and regulatory mechanisms within maize and across different species (Figure 5). Through synteny analysis, 15 syntenic gene pairs were identified among the 34 PHT gene family members on the 10 chromosomes of maize, specifically: *ZmPHT1-2*/*ZmPHT1-4*, *ZmPHT1-4*/*ZmPHT1-8*, *ZmPHT4-1*/*ZmPHT4-3*, *ZmPHT1-2*/*ZmPHT1-8*, *ZmPHT1-11*/*ZmPHT1-5*, *ZmPHT5-1*/*ZmPHT5-2*, *ZmPHT4-7*/*ZmPHT4-8*, *ZmPHT4-2*/*ZmPHT4-9*, *ZmPHO1-H2*/*ZmPHO1-H3*, *ZmPHT3-3*/*ZmPHT3-4*, *ZmPHT3-3*/*ZmPHT3-5*, *ZmPHT3-4*/*ZmPHT3-5*, *ZmPHT4-8*/*ZmPHT4-6*, and *ZmPHT5-3*/*ZmPHT5-4* (Figure 5a). Notably, the *ZmPHT5-2* gene formed a syntenic pair with *Zm00001eb428590*, which was not identified as a PHT family member. This suggests the possibility of gene function loss or the loss of conserved regions during gene duplication or genomic rearrangement. Therefore, selective pressure on the syntenic gene pairs was assessed, including non-synonymous mutations (Ka), synonymous mutations (Ks), and their ratio (Ka/Ks). The overall Ks values of the syntenic pairs ranged from 0.16 to 2.20, with the *ZmPHT3-3*/*ZmPHT3-5* pair having the highest value, indicating that this gene pair diverged a long time ago (Appendix A). The Ka/Ks values of all duplicated gene pairs were less than 1.00, ranging from 0.03 to 0.41, indicating that these gene pairs have undergone strong purifying selection during evolution, suggesting that their functions are conserved. Synteny analysis was also performed on maize B73, the phosphorus-tolerant line Mo17, and sorghum (*Sorghum bicolor*). The synteny analysis between the genomes of maize lines B73 and Mo17 identified 51 syntenic pairs, indicating that despite differences in phosphorus use efficiency, these PHT genes remain highly conserved throughout evolution (Figure 5b). Furthermore, specific gene variations in Mo17 may contribute to its unique phosphorus tolerance. Comparative analysis with the sorghum genome identified 37 syntenic pairs between the two species (Figure 5c). This further revealed the conservation and functional divergence of these PHT genes between species. The syntenic regions in the sorghum genome also suggest that PHT genes may share common regulatory mechanisms across these species.

### 2.6. Transcriptome and Enrichment Analysis of ZmPHT Genes

By analyzing RNA-seq data from various developmental stages of maize organs, we can understand the expression patterns of ZmPHT gene family members in different tissues. Specific samples include Primary_Root_6_days_after_sowing_GH(PR_6d_GH), Whole_Primary_Root_7d (WPR_7d), and Secondary_Root_7_8_Days (SR_7-8d). Additionally, there are Heat_treated_seedlings (Heat_Seedlings) and Salt_treated_seedlings (Salt_Seedlings). Leaf samples from various growth stages include V3_Topmost_leaf (V3_Top_Leaf), V5_Tip_of_Stage_2_Leaf (V5_Tip_Leaf), and V7_Tip_of_transition_leaf (V7_Tip_Trans_Leaf) (Figure 6a and Appendix A). These data analyses allowed us to gain deeper insights into the functions of ZmPHT gene family members in maize growth and environmental responses. The expression analysis revealed that *ZmPHT3-4* and *ZmPHO1-H3* showed notably high expression levels in root tissues compared to other genes in their respective subfamilies (Figure 6a). Their expression in root tissues suggests potential roles in phosphate uptake and transport. The high expression of *ZmPHT4-2*, *ZmPHT3-4*, *ZmPHT4-4*, and *ZmPHT4-5* under stress indicates their potential key role in adapting to environmental stresses. *ZmPHT2-1*, *ZmPHT3-4*, *ZmPHT4-2*, *ZmPHT4-4*, *ZmPHT4-5*, *ZmPHT4-9*, and *ZmPHT5-4* also exhibited high expression in leaves during different growth stages, indicating that these genes may be involved in photosynthesis and nutrient utilization. Furthermore, *ZmPHT3-4*, *ZmPHT4-2*, and *ZmPHT4-1* also showed high expression in different functional regions of the leaves (e.g., LZ1_Symm, LZ2_Stoma, and LZ3_Growth), highlighting their crucial roles in leaf development and function. On the other hand, *ZmPHT1-7*, *ZmPHT1-11*, *ZmPHT1-9*, *ZmPHT1-6*, *ZmPHT1-10*, *ZmPHT3-3*, *ZmPHT4-3*, and *ZmPHT1-1* showed generally low expression across various tissues and developmental stages, suggesting that these genes may have limited functions under normal conditions or may only be activated under specific environmental or physiological conditions.

Based on our previous research, extreme phosphorus use efficiency lines, including the low P-sensitive line P1 and low P-tolerant line P2, were grown under both low-phosphate (LP) and normal-phosphate (NP) conditions. First, transcriptome data from PHT gene family members were subjected to principal component analysis (PCA) (Figure 6b). The results revealed a clear separation between the two lines under different phosphorus treatments, reflecting their contrasting phosphorus use efficiency. These differences highlighted the distinct transcriptional characteristics of the PHT gene family under low- and moderate-phosphorus conditions. *ZmPHT1-3*, *ZmPHT1-10*, *ZmPHT1-4*, and *ZmPHO1-H3* showed increased expression in both low P-tolerant and low P-sensitive maize lines (Figure 6c and Appendix A). This suggests that these ZmPHT genes may play key roles in different lines, particularly in phosphorus stress response mechanisms. The general increase in the expression of these genes indicates their involvement in the response to phosphorus-deficient environments and may represent a general mechanism for plant adaptation to low-P conditions. This further supports the importance of these genes in maintaining phosphate homeostasis in plants, with *ZmPHT1-3* showing significantly higher expression in the low P-tolerant line compared to the low P-sensitive line, suggesting that this gene may play a key role in phosphorus stress tolerance, as its high expression may be related to efficient phosphorus utilization.

GO enrichment analysis of the ZmPHT gene family revealed significant enrichment in the categories ‘inorganic phosphate transmembrane transporter activity’ and ‘phosphate ion transport’, indicating that the ZmPHT gene family plays a key role in regulating plant responses to phosphorus stress (Figure 6d). This analysis also found that the ZmPHT gene family is closely related to mitochondrial membrane and inner membrane functions, highlighting their role in maintaining phosphate homeostasis and energy metabolism in plant cells. Furthermore, a protein–protein interaction network of ZmPHT gene family members was constructed to explore potential interactions between proteins and their functional associations in phosphate uptake and metabolism (Figure 6e). The figure shows the interactions among 34 PHT family members, with *ZmPHT2-1* serving as the central node, exhibiting strong interactions with many other PHT proteins, highlighting its central role in the network. Additionally, the connections in the network reveal complex functional relationships between these proteins, which may reflect their cooperative roles in regulating plant phosphorus status.

In summary, the expression of the ZmPHT gene family is closely related to plant adaptive responses to phosphorus-deficient stress, providing a theoretical basis for the further exploration of molecular mechanisms to improve crop phosphorus use efficiency.

## 3. Discussion

### 3.1. Systematic Identification and Phylogenetic Analysis Reveal Functional Divergence in the Maize PHT Gene Family

This study systematically identified 34 members of the maize PHT gene family, categorized into six subfamilies, and performed detailed analyses of their structural characteristics, chromosomal distribution, physicochemical properties, and transmembrane helices. The results show substantial differences in amino acid length, molecular weight, and transmembrane helix structure among the PHT subfamilies, with the PHT1 subfamily likely playing a key role in phosphate transport. Phylogenetic tree analysis revealed the evolutionary relationship of maize PHT genes with those in monocots such as sorghum and rice, indicating functional divergence and adaptive evolution in monocot plants. Additionally, the protein instability index and isoelectric point analysis of PHT family members suggest that they may have diverse roles in the dynamic regulation and maintenance of cellular phosphate levels [39,40].

Building on previous studies, this research not only fills the gap in the study of the PHT2, PHT3, PHT4, PHT5, and PHO1 subfamilies in maize, but also underlines the importance of the PHT1 family in phosphate absorption in plants [37]. In summary, unlike previous studies that focused on the PHT1 gene family and its association with arbuscular mycorrhizal fungi, our research provides a comprehensive analysis of all PHT subfamilies in maize, integrating transcriptomic data and protein–protein interaction networks. This study offers new insights into the regulatory mechanisms of the PHT gene family in response to phosphorus stress. Previous studies have shown that PHT1 proteins can effectively enhance phosphate absorption efficiency under low-phosphate stress and play an important role in plants such as wheat and *Salvia miltiorrhiza* [44,45,46]. This study, through phylogenetic tree analysis and gene identification, further reveals the structural and functional differentiation of the maize PHT gene family, as well as its regulatory network in phosphate transport, signal transduction, and stress response, providing a theoretical basis for future in-depth exploration of the gene family’s functions.

The subcellular localization prediction and intron number characteristics of the ZmPHT gene family collectively reveal its functional differentiation and environmental adaptability. Most members of the ZmPHT gene family are localized to the plasma membrane, suggesting their primary roles in phosphate uptake and intercellular transport. In contrast, the PHT3 subfamily and some members of the PHT4 subfamily exhibit diverse localizations, including the cytoplasm, chloroplast, nucleus, and extracellular space, indicating their potential roles in phosphate metabolism regulation, signal transduction, and organelle-specific functions. Subcellular localization is known to regulate protein–protein interactions, thereby optimizing protein functionality [47]. This pattern underscores the functional coordination of the ZmPHT gene family in phosphate utilization and transport, providing a basis for further elucidating its functional mechanisms and breeding phosphate-efficient crops. Moreover, introns, as a crucial component of eukaryotic genomes, also play an important role in the functional differentiation of the ZmPHT gene family. Studies have shown that introns impose an energetic burden on cells, and an excessive number of introns may disrupt cellular functions [48,49]. However, a reduced intron number can enhance the environmental adaptability of plants [50]. In the ZmPHT gene family, different subfamilies exhibit diverse intron number characteristics [51]. The PHT1 and PHT2 subfamilies have fewer introns, suggesting that their streamlined gene structures facilitate rapid responses to phosphate stress and efficient phosphate uptake. Conversely, the PHT3, PHT4, PHT5, and PHO1 subfamilies possess more introns, indicating their potential involvement in complex phosphate transport, distribution processes, and more refined transcriptional regulation. This diversity in intron numbers highlights the functional specialization of the ZmPHT gene family in phosphate uptake, transport, and homeostasis regulation, providing critical insights into the mechanisms of plant adaptation to phosphate stress.

### 3.2. Promoter Region and Synteny Analysis of Maize PHT Genes Reveals Potential Signal Regulation and Gene Conservation

This study analyzed the upstream promoter regions of maize PHT genes and revealed their potential functions in hormone signaling, stress responses, and light signal regulation. The promoter regions are rich in cis-regulatory elements such as ABREs, AREs, and G-box, indicating that this gene family may play a key role in phosphate uptake, stress adaptation, and light response. The promoter regions of the PHT1 subfamily exhibit significant hormone regulation potential, especially *ZmPHT1-4* and *ZmPHT1-5*, which are enriched with elements related to abscisic acid (ABA), jasmonic acid (JA), and gibberellin (GA). The results suggest that these genes may regulate high expression of phosphate transport proteins under hormone regulation and environmental stress, promoting plant adaptation to low-phosphate environments [16,18]. Additionally, the promoter regions of the PHO1 subfamily are enriched with hormone- and stress-related elements, especially in *PHO1-H1*, where a large number of LTR and GC motif elements were detected, suggesting its potential role in plant stress signal transduction, consistent with previous studies [26,27,28,31].

Maize collinearity analysis identified 15 pairs of collinear genes, among which *ZmPHT5-2* formed a collinear pair with a non-PHT family gene (*Zm00001eb428590*), suggesting potential loss of function or conserved regions during gene duplication or genome rearrangement. Selection pressure analysis showed that the Ka/Ks ratios of all gene pairs were less than 1.00, indicating that these genes have undergone strong purifying selection, are functionally conserved, and may play stable roles in phosphate transport and utilization in plants.

Upstream promoter regions analysis revealed the potential functions of PHT genes in hormone signaling and stress responses, while collinearity analysis demonstrated the conservation of the PHT gene family across different crops, providing a reference for future studies on candidate genes related to phosphate stress responses.

### 3.3. Transcriptome Data Analysis Reveals Diverse Functions of Maize PHT Genes in Phosphate Uptake and Stress Response

RNA-seq data from maize in different developmental stages and stress conditions were analyzed to reveal the diverse functions of the PHT gene family in phosphate uptake and growth. Transcriptome analysis showed that ZmPHT genes exhibit specific expression patterns in different tissues, developmental stages, and environmental conditions, supporting their diverse roles in phosphate metabolism and stress response. High expression of *ZmPHT3-4* and *ZmPHO1-H3* in root tissues suggests that they may be involved in phosphate uptake and primary transport, particularly in maintaining stable phosphate levels under low-phosphate conditions. This result is consistent with previous studies, which showed that high expression of PHT genes in the roots is closely related to phosphate uptake, as roots are the primary organs for phosphate acquisition, where these genes may play critical roles [16,24].

The high expression of *ZmPHT3-4*, *ZmPHT4-2*, and Zm*PHT4-1* in leaves suggests that they may be involved in phosphate redistribution during photosynthesis. Additionally, *ZmPHT3-4*, *ZmPHT4-2*, and *ZmPHT4-1* show high expression in different functional areas of the leaf (e.g., LZ1_Symm, LZ2_Stoma, LZ3_Growth), highlighting their importance in leaf development and function. Previous studies have shown that the *PHT4-4* gene can prevent the downregulation of photosynthesis genes, leading to the stay-green phenotype under iron–phosphorus deficiency, which supports the regulation of photosynthesis under such stress conditions [52].

Under various stresses, *ZmPHT4-2*, *ZmPHT3-4*, *ZmPHT4-4*, and *ZmPHT4-5* exhibit high expression, indicating their potential regulatory roles under stress, particularly in plant phosphate uptake and utilization. This result is consistent with previous research, which showed that PHT family members play adaptive regulatory roles under various stresses, particularly phosphate deficiency, salt stress, and drought stress [53,54,55].

By analyzing the transcriptomic data of different phosphorus-tolerant lines under low- and normal-phosphorus stresses, principal component analysis (PCA) showed a clear separation in gene expression between phosphorus-tolerant and phosphorus-sensitive lines under low-phosphorus stress and normal-phosphorus conditions, further validating the critical role of PHT family genes in response to phosphorus stress. Specifically, the upregulation of *ZmPHT1-3*, *ZmPHT1-10*, *ZmPHT1-4*, and *ZmPHO1-H3* in phosphate-tolerant lines suggests that these genes may play central roles in phosphate stress tolerance, enhancing plant adaptability under low-phosphate stress. *ZmPHT1-4* has one homolog gene in rice, *OsPHT1-8*, which was identified through experimental studies as a key regulator of phosphate starvation responses in rice [56]. GO enrichment analysis showed significant enrichment of the ZmPHT gene family in ‘inorganic phosphate transmembrane transporter activity’ and ‘phosphate ion transport’, further confirming their key roles in phosphate transport and homeostasis. Transcriptomic analysis has laid the foundation for further investigation of the regulatory mechanisms of ZmPHT genes in phosphorus use efficiency, phosphorus stress adaptation, and growth and development.

### 3.4. Evolutionary Adaptation and Functional Divergence of the PHT Gene Family

The PHT gene family strikes a balance between conservation and diversity, reflecting the dual evolutionary pressures of maintaining phosphate absorption mechanisms and adapting to species-specific ecological needs.

On a macroevolutionary scale, monocots such as maize, sorghum, and rice show close evolutionary relationships, supported by the conserved distribution of PHT gene subfamilies, especially the dominance of PHT1 [57,58]. This highlights the essential role of PHT1 genes in phosphate uptake. In contrast, dicots like tomato and white lupin display unique adaptations. Tomato significantly upregulates its PHT1 genes during arbuscular mycorrhizal symbiosis, emphasizing its strategy to enhance phosphate uptake through fungal interactions [59]. White lupin, on the other hand, follows a distinct evolutionary path, with an expanded PHT2 subfamily that supports its specialized cluster root system, enabling efficient phosphate solubilization and absorption in nutrient-poor soils [60]. At the molecular level, PHT proteins consistently show high aliphatic indices and positive hydropathicity values. These features indicate that their transmembrane domains have been finely tuned for stable integration into lipid bilayers, ensuring efficient phosphate transport. In summary, the evolution of the PHT gene family demonstrates how plants adaptively regulate genetic mechanisms to overcome phosphate availability challenges. These insights not only enhance our understanding of plant adaptation but also provide opportunities to improve phosphorus use efficiency in crops, paving the way for more sustainable agricultural practices.

## 4. Materials and Methods

### 4.1. Identification and Characterization of PHT Family Genes

In this study, genomic, annotation, and protein sequence data for *Arabidopsis thaliana* and *Zea mays* were first obtained from the Ensembl Plants database. The Arabidopsis data were sourced from version TAIR10, while the maize data were selected from version Zm-B73-REFERENCE-NAM-5.0. A protein sequence database for maize was constructed using the makeblastdb function in BLAST+ v2.14.0 software, and the blastp function was employed to compare the PHT protein sequences of Arabidopsis, which were retrieved from the TAIR genome database (https://www.arabidopsis.org, accessed on 11 January 2024), with those of maize, setting the alignment parameters to output format 6 and an E-value of 1 × 10^−5^ [60]. To ensure high-quality alignment results, alignments with a similarity greater than 30% were filtered out for subsequent analysis. Further validation was conducted through the GenomeNet (https://www.genome.jp, accessed on 11 January 2024) and SMART (http://smart.embl-heidelberg.de/help/smart_glossary.shtml, accessed on 11 January 2024) websites. In analyzing the PHT1 subfamily, we referenced a previous study based on an earlier genome version, which identified 13 PHT1 subfamily genes. However, due to updates in the genome version and changes in gene annotation, directly using the gene nomenclature from previous studies may lead to confusion and inaccuracies. After converting these gene sequences to the latest version, Zm-B73-REFERENCE-NAM-5.0, it was found that two genes (*ZmPHT1-11* and *ZmPHT1-12*) were merged into a single gene in the new version. Additionally, some genes originally located on chromosome 5 were absent in the latest genome version. Through BLAST comparative analysis, it was determined that these genes may have redundancy or ambiguous correspondence with other genes in the new version. Given the aforementioned reasons, to ensure the accuracy and consistency of gene naming in this study, a re-identification and renaming of the PHT1 subfamily genes were conducted based on the Zm-B73-REFERENCE-NAM-5.0 version. A unified gene numbering system was adopted to avoid confusion arising from differences in genome versions (Appendix A). This approach enhanced the reliability of the findings and provided a clear reference for future research. Chromosomal localization was visualized using TBtools v2.056 software, while the physicochemical properties of the ZmPHT proteins, such as the theoretical isoelectric point (pI) and relative molecular weight (MW), were assessed through the EXPASY (https://web.expasy.org/compute_pi/, accessed on 11 January 2024) website. Subcellular localization predictions were made using the WoLF PSORT website (https://wolfpsort.hgc.jp, accessed on 11 January 2024), and transmembrane domain predictions were conducted via the TMHMM-2.0 website (https://services.healthtech.dtu.dk/services/TMHMM-2.0/, accessed on 11 January 2024). The phylogenetic tree was constructed by first performing a multiple sequence alignment of all PHT protein sequences using MAFFT v7.487 software [61] with global pair settings and a maximum of 1000 iterations. Subsequently, the Neighbor Joining (NJ) method from Treebest v1.9.2 software was employed along with the Jones–Taylor–Thornton (JTT) substitution model, and 1000 bootstrap replications were conducted to generate the final phylogenetic tree file.

### 4.2. Gene Structural Analysis of PHT Genes

To effectively analyze gene structure and domains, protein sequences were first submitted to the MEME Suite website (https://meme-suite.org/meme/, accessed on 11 January 2024). Gene structure annotations in GTF/GFF3 format for maize were downloaded from the Ensembl plant database (https://plants.ensembl.org/info/data/ftp/index.html, accessed on 11 January 2024). Protein domain predictions were then performed using the NCBI CDD tool (https://www.ncbi.nlm.nih.gov/Structure/cdd/wrpsb.cgi, accessed on 11 January 2024). Next, the TBtools software was used for the visualization of motifs and gene structures. In addition, to analyze cis-regulatory elements in the promoter regions of ZmPHT genes, a 2000 bp sequence upstream of the transcription start codon (ATG) for each PHT genes was extracted. These sequences were then submitted to the PlantCARE database (https://bioinformatics.psb.ugent.be/webtools/plantcare/html/, accessed on 11 January 2024) for prediction.

### 4.3. Chromosomal Distribution and Gene Duplication Events

To investigate the genomic synteny relationships between maize variety B73 (*Zea mays* L. B73), maize variety Mo17 (*Zea mays* L. Mo17), and sorghum (*Sorghum bicolor*), synteny analysis was performed using the One Step McScanX-Super Fast module in the TBtools software [62]. The genome sequence files and gene structure annotation files of the three species were imported into the software, with the e-value set to 1 × 10^−5^ and the number of Blast hits set to 10. After completing the synteny analysis, the results were further visualized using the Dual Synteny Plot module in TBtools. The maize genome synteny Circos plot was visualized using the Advanced Circos module. The calculation of Ka, Ks, and Ka/Ks values was performed using the Simple Ka/Ks Calculator module in TBtools.

### 4.4. Transcriptome Data Acquisition and Analysis Methods for ZmPHT Genes

Transcriptome data related to phosphorus stress in maize tissues were downloaded from the Qteller website (https://qteller.maizegdb.org, accessed on 11 January 2024). Based on previous research from our group, we selected the Shangzhuang Experimental Station of China Agricultural University (referred to as SZ) to conduct long-term field trials, with a focus on the effects of low-phosphorus and normal-phosphorus treatments. Since 1985, no phosphorus fertilizer has been applied to the low-phosphorus area (P0) of the experimental field, while 45 kg/ha of P_2_O_5_ has been applied annually before planting in the normal-phosphorus area (P1). Both areas received 240 kg/ha of total nitrogen, with all other management practices kept consistent. The experimental field measured 72 m in length and 15 m in width. In these two test fields, two extreme lines identified in previous studies by our research group were planted, designated as P1 (phosphorus-sensitive line) and P2 (phosphorus-tolerant line). At the five-leaf stage of maize, the above-ground parts of the plants were sampled, and three replicate experiments were performed. The samples were immediately frozen in liquid nitrogen after collection and stored in a −80 °C freezer. The experimental design included two treatments (low phosphorus and normal phosphorus), two genotypes, and three replicates, totaling 12 samples. After grinding the whole plants, an appropriate amount of sample was taken for subsequent analysis. In the subsequent transcriptome data analysis, gene length information and expression data were first retrieved. Then, based on the expression levels and lengths of each gene, the *FPKM* value for each gene in each sample was calculated. Subsequently, *TPM* (transcripts per million) was calculated by normalizing the *FPKM* values. To facilitate further analysis, duplicate numbers in sample names were removed, and the average *TPM* value for each gene in each tissue section was calculated.

The calculation formula for *FPKM* (fragments per kilobase of transcript per million mapped reads) is$$FPKM=\frac{count}{genelength}÷\frac{totalcounts}{1000000}$$

*count*: the number of reads mapped to a specific gene in a given sample.

*gene length*: the length of the gene (in base pairs).

*total counts*: the total number of reads for all genes in the sample.

The calculation formula for *TPM* (transcripts per million) is$$TPM=\frac{FPKM}{\sum FPKM}\times 1000000$$

*FPKM*: the *FPKM* value calculated previously.

∑FPKM is the sum of *FPKM* values for all genes in the sample.

Additionally, we used three R packages—biomaRt v2.62.0, enrichplot v 1.26.2, and clusterProfiler v 4.14.3 [63,64]—for GO enrichment analysis. We connected to the Ensembl Plant database [65] using biomaRt to obtain GO annotation information for maize genes. Then, we performed GO enrichment analysis on the gene lists read from external files using the clusterProfiler v 4.14.3package and adjusted the *p*-values using the FDR method to ensure result accuracy. Finally, we visualized the enrichment analysis results using Python.

## 5. Conclusions

This study systematically identified and analyzed 34 members of the maize PHT gene family, revealing their potential roles in phosphate uptake, transport, and stress response, particularly under low-phosphorus (P) stress. Through phylogenetic analysis, promoter cis-elements, synteny, and transcriptome analysis, potential phosphate stress-related genes were identified, particularly *ZmPHT1-3*, *ZmPHT1-10*, *ZmPHT1-4*, and *ZmPHO1-H3*, which showed prominent roles in plant responses to phosphate deficiency. These research findings provide key candidate genes for improving phosphorus use efficiency and crop stress tolerance through molecular breeding and have significant application potential in future functional validation and crop improvement.

## Figures and Tables

**Figure 1 ijms-26-01445-f001:**
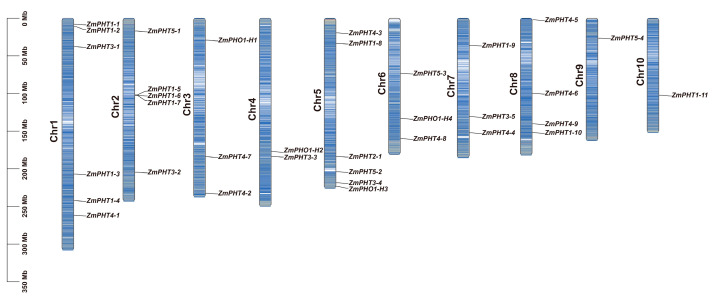
Chromosomal localization of ZmPHT genes in maize. The positions of ZmPHT gene members are marked on each chromosome. The blue bands on the chromosomes represent gene density, with darker colors indicating higher gene density in that region.

**Figure 2 ijms-26-01445-f002:**
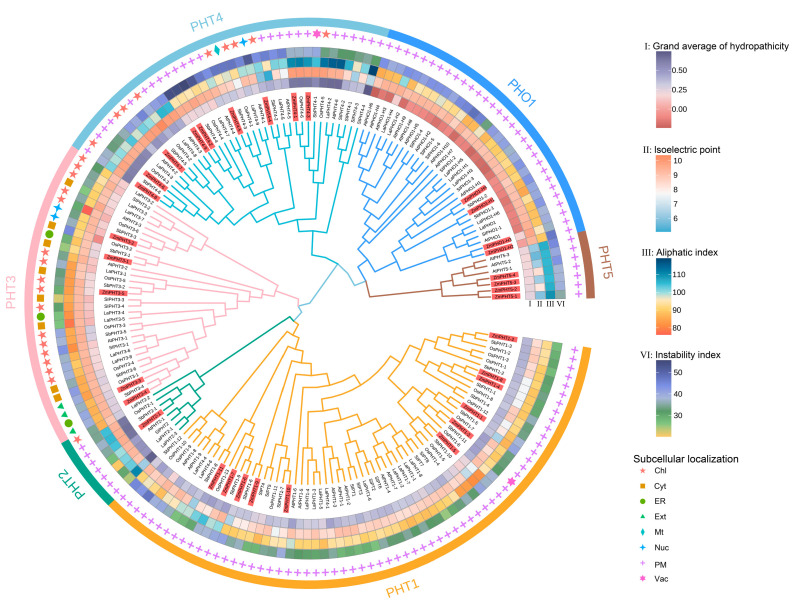
Phylogenetic tree of PHT family members across different species. The tree was constructed based on the PHT protein sequences from maize, rice, Arabidopsis, tomato, white lupin, and sorghum. The outer circular annotations display the physicochemical properties of PHT proteins, including the grand averages of hydropathicity (I), isoelectric point (II), aliphatic index (III), and instability index (IV). Subcellular localization predictions are represented with distinct icons indicating the chloroplast (Chl), cytoplasm (Cyt), endoplasmic reticulum (ER), extracellular space (Ext), mitochondria (Mt), nucleus (Nuc), plasma membrane (PM), and vacuole (Vac). The outermost ring groups the PHT subfamily members. Different colored lines represent different PHT subfamilies.

**Figure 3 ijms-26-01445-f003:**
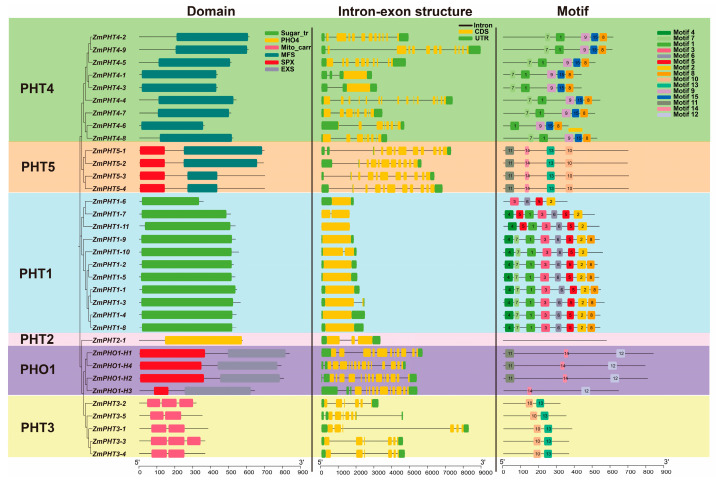
Structural analysis of the ZmPHT gene family in maize. The upper-right corner of each panel shows the legend corresponding to the figure, and the scale at the bottom represents the length of the protein sequence or gene sequence for each ZmPHT family member. Numbers within the colored boxes represent different conserved motifs, as indicated in the legend on the right.

**Figure 4 ijms-26-01445-f004:**
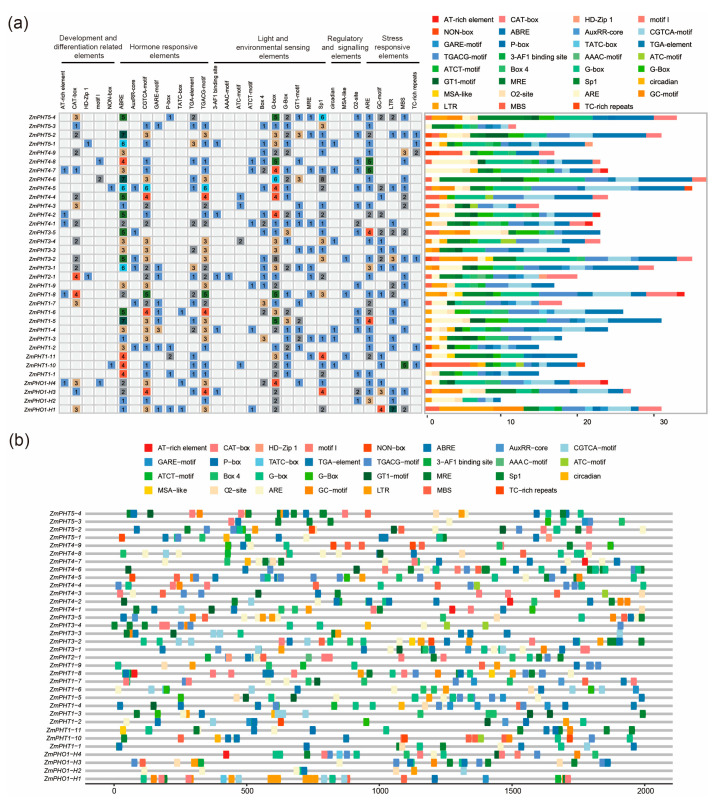
Cis-regulatory element analysis of ZmPHT genes in maize. (**a**) The number of different types of cis-regulatory elements in the promoter sequences of ZmPHT genes is shown, including elements related to growth and development, hormone response, light regulation, signal transduction, and stress response. The bar chart shows the number of cis-elements, with similar color schemes representing the same functional categories, such as shades of red representing growth and development, blue shades representing hormone response, green shades representing light and environmental sensing, beige shades representing the regulation of signaling elements, and orange shades representing stress response. (**b**) Cis-regulatory element distribution in the ZmPHT genes. The scale at the bottom represents the 2000 bp upstream sequence for each ZmPHT gene.

**Figure 5 ijms-26-01445-f005:**
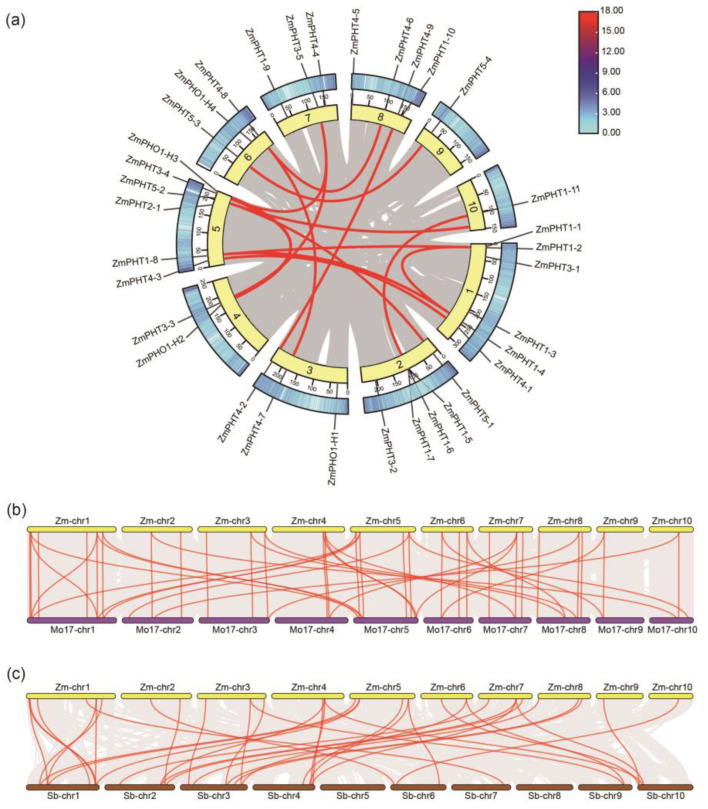
Synteny analysis of PHT gene families in different plants. (**a**) Synteny Circos plot within the maize genome. The inner ring indicates chromosome numbers, and the bar charts outside the chromosomes display gene density, with darker colors representing higher gene density. The numbers in the yellow ring represent maize chromosome numbers, while the red connecting lines indicate syntenic gene pairs. (**b**) Synteny plot between maize lines B73 and Mo17. Yellow lines represent the chromosomes of the B73 line, purple lines represent the chromosomes of the Mo17 line, and the red connecting lines show syntenic gene pairs between the two maize inbred lines. (**c**) Synteny plot between maize and sorghum chromosomes. Yellow lines represent the chromosomes of the maize B73 line, brown lines represent the chromosomes of sorghum, and the red connecting lines indicate syntenic gene pairs between the two species.

**Figure 6 ijms-26-01445-f006:**
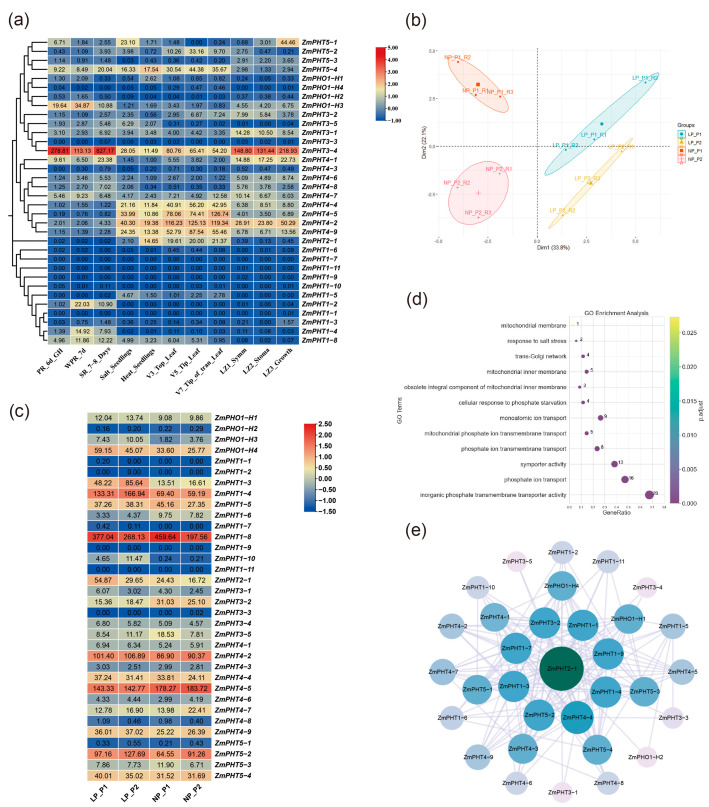
Expression patterns, GO enrichment, and protein interaction network of PHT family members. (**a**) Expression profile analysis based on *FPKM* RNA-seq data. PR_6d_GH: Primary_Root_6_days_after_sowing_GH; WPR_7d: Whole_Primary_Root_7d; SR_7-8d: Secondary_Root_7_8_Days; Salt_Seedlings: Salt_treated_seedlings; Heat_Seedlings: heat-treated seedling; V3_Top_Leaf: V3_Topmost_leaf; V5_Tip_Leaf: V5_Tip_of_Stage_2_Leaf; V7_Tip_Trans_Leaf: V7_Tip_of_transition_leaf; LZ1_Symm: Leaf_Zone_1_Symmetrical; LZ2_Stoma: Leaf_Zone_2_Stomatal; LZ3_Growth: Leaf_Zone_3_Growth. The numbers in the heatmaps represent *TPM* values. (**b**) Transcriptome principal component analysis (PCA) of ZmPHT genes family members, with a PCA plot showing four groups: LP-P1, LP-P2, NP-P1, and NP-P2. Principal Component 1 (Dim 1) and Principal Component 2 (Dim 2) account for the majority of the variation in the data. (**c**) Expression profile analysis based on *FPKM* RNA-seq data. LP: low-phosphorus condition; NP: normal-phosphorus condition; P1: phosphorus-sensitive line; P2: phosphorus-tolerant line. The numbers in the heatmaps represent *TPM* values. (**d**) GO enrichment pathways of PHT genes. (**e**) Protein–protein interaction (PPI) network of PHT family members. The central node is the *ZmPHT2-1* gene. The connections represent interactions between genes.

**Table 1 ijms-26-01445-t001:** Distribution of PHT gene family members across different species.

Species	Total PHT Genes	PHT1	PHT2	PHT3	PHT4	PHT5	PHO1
*Zea mays*	34	11	1	5	9	4	4
*Oryza sativa*	26	13	1	6	6	-	-
*Arabidopsis thaliana*	33	9	1	3	6	3	11
*Solanum lycopersicum*	23	8	1	4	4	-	6
*Lupinus albus*	35	8	3	8	9	-	7
*Sorghum bicolo* *r*	27	12	1	6	6	-	2

- indicates that no relevant gene numbers were found in the referenced studies.

## Data Availability

All relevant data are included within the paper and its Appendix A.

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
