# Peer review of "Genome-Wide Identification and Expression Analysis of the Phosphate Transporter Gene Family in Zea mays Under Phosphorus Stress"

_ijms, 2025, doi:10.3390/ijms26041445_

Round 1
Reviewer 1 Report
Comments and Suggestions for Authors
The manuscript entitled "Genome-Wide Identification and Expression Analysis of the Phosphate Transporter (PHT) Gene Family in Zea mays under Phosphorus Stress" is a well-structured and comprehensive study, providing valuable insights into the identification, characterization, and expression profiling of the PHT gene family in maize under phosphorus stress. Phosphorus is a critical nutrient, and improving phosphorus use efficiency (PUE) is essential for enhancing crop productivity and sustainability. This study systematically identifies PHT genes, conducts phylogenetic and transcriptome analyses, and highlights key candidate genes (ZmPHT1-3, ZmPHT1-4, ZmPHT1-10, and ZmPHO1-H3) with potential roles in phosphorus stress tolerance. The inclusion of cis-regulatory element analysis further strengthens the manuscript by linking gene function to stress responses and hormonal regulation. As the first comprehensive analysis of the PHT gene family in maize, it offers valuable molecular resources for breeding phosphorus-efficient maize varieties and advancing sustainable agriculture. Minor improvements, including a general English review to address grammatical and typographical errors, ensuring keywords are in alphabetical order, italicizing gene names, using plain text for protein names, and appropriately linking figures, tables, and references to the text, will further enhance the manuscript's clarity and presentation. This work makes a significant contribution to the understanding of phosphorus stress in maize and provides a strong foundation for future research and crop improvement efforts. The work is welldone and I suggest the team to continue the current work and further deepen the understanding of which potential PHT involve phosphorus use efficiency. Finally I accept this manuscript after these minor revision.
Please find the detail comments: This study addresses how maize adapts to low-phosphorus stress and identifies key members of the PHT gene family involved in improving phosphorus use efficiency (PUE). This question is highly relevant to modern agriculture, where phosphorus availability is a major constraint on crop productivity. By systematically characterizing the PHT gene family, the research provides valuable insights into the genetic and molecular mechanisms underlying phosphorus uptake and utilization in maize. The topic is original and significant, filling a critical gap in understanding maize's response to phosphorus stress. As the first comprehensive analysis of the PHT gene family in maize, the study offers practical implications for enhancing phosphorus efficiency, which is vital for sustainable agriculture. By focusing on genes such as ZmPHT1-3, ZmPHT1-4, and ZmPHO1-H3, it identifies key molecular targets for breeding phosphorus-efficient maize varieties. Methodological improvements could enhance its impact. Functional validation of identified genes through knockout or overexpression studies would strengthen conclusions. Including physiological data, such as phosphorus uptake rates or yield performance under stress, would provide practical insights. Comparative analyses across other crops could validate the broader applicability of the findings. The conclusions align with the evidence, linking the identified PHT genes with their potential to improve PUE. However, experimental validation under field conditions would bolster their applicability. Figures are of low quality images and can be improved. Addressing the suggested refinements would further enhance its value and utility.

Author Response
Comment 1:
The manuscript entitled "Genome-Wide Identification and Expression Analysis of the Phosphate Transporter (PHT) Gene Family in Zea mays under Phosphorus Stress" is a well-structured and comprehensive study, providing valuable insights into the identification, characterization, and expression profiling of the PHT gene family in maize under phosphorus stress. Phosphorus is a critical nutrient, and improving phosphorus use efficiency (PUE) is essential for enhancing crop productivity and sustainability. This study systematically identifies PHT genes, conducts phylogenetic and transcriptome analyses, and highlights key candidate genes (ZmPHT1-3, ZmPHT1-4, ZmPHT1-10, and ZmPHO1-H3) with potential roles in phosphorus stress tolerance. The inclusion of cis-regulatory element analysis further strengthens the manuscript by linking gene function to stress responses and hormonal regulation. As the first comprehensive analysis of the PHT gene family in maize, it offers valuable molecular resources for breeding phosphorus-efficient maize varieties and advancing sustainable agriculture.
Response 1:
Thank you for your efforts and positive comments!
Comment 2:
Minor improvements, including a general English review to address grammatical and typographical errors
Response 2:
Following your comment, we have carefully revised the manuscript to address grammatical and typographical errors, ensuring the language is clear and accurate.
Comment 3:
ensuring keywords are in alphabetical order, italicizing gene names
Response 3:
Thank you for pointing this out. We agree with your comment. The keywords have been rearranged in alphabetical order for better consistency. The ‘PHT’ in ‘PHT family’ is commonly written in plain text and not in italic. The updated text can be found on Line [29] in the revised manuscript.
Comment 4:using plain text for protein names
Response 4:
Thank you for your suggestion. Regarding the use of italics for gene names, we have revised the text to ensure singular gene names are italicized, while plural forms representing gene families remain in plain text, as per the convention.
Comment 5:
appropriately linking figures, tables, and references to the text, will further enhance the manuscript's clarity and presentation.
Response 5:
Thank you for your suggestion. All figures, tables, and references have been re-checked and appropriately linked to the text for consistency and accuracy.
Comment 6:
This work makes a significant contribution to the understanding of phosphorus stress in maize and provides a strong foundation for future research and crop improvement efforts. The work is welldone and I suggest the team to continue the current work and further deepen the understanding of which potential PHT involve phosphorus use efficiency. Finally, I accept this manuscript after these minor revision.
Response 6:
Thank you again for your comments and suggestions!
Reviewer 1:
Comment 1:
Please find the detail comments: This study addresses how maize adapts to low-phosphorus stress and identifies key members of the PHT gene family involved in improving phosphorus use efficiency (PUE). This question is highly relevant to modern agriculture, where phosphorus availability is a major constraint on crop productivity. By systematically characterizing the PHT gene family, the research provides valuable insights into the genetic and molecular mechanisms underlying phosphorus uptake and utilization in maize. The topic is original and significant, filling a critical gap in understanding maize's response to phosphorus stress. As the first comprehensive analysis of the PHT gene family in maize, the study offers practical implications for enhancing phosphorus efficiency, which is vital for sustainable agriculture. By focusing on genes such as ZmPHT1-3, ZmPHT1-4, and ZmPHO1-H3, it identifies key molecular targets for breeding phosphorus-efficient maize varieties. Methodological improvements could enhance its impact. Functional validation of identified genes through knockout or overexpression studies would strengthen conclusions. Including physiological data, such as phosphorus uptake rates or yield performance under stress, would provide practical insights. Comparative analyses across other crops could validate the broader applicability of the findings. The conclusions align with the evidence, linking the identified PHT genes with their potential to improve PUE.
Response 1:
Thank you for your efforts and positive comments!
Comment 2:
However, experimental validation under field conditions would bolster their applicability.
Response 2:
Thank you for your valuable suggestion. Just as you suggested, we are going to validate those promising functional genes through molecular experiments, physiological measurement, and field trials. And we’ll publish those consequent studies in other stories. Thanks a lot!
Comment 3:
Figures are of low quality images and can be improved. Addressing the suggested refinements would further enhance its value and utility.
Response 3:
Thank you for your comment. The images appeared to be of lower quality due to file compression during the upload process. To ensure clarity, we have provided the original high-quality images in the "High-quality images" folder for your reference at any time.
Reviewer 2 Report
Comments and Suggestions for Authors
Title: Genome-Wide Identification and Expression Analysis of the Phosphate Transporter (PHT) Gene Family in Zea mays under Phosphorus Stress
Summary: The authors identified the Phosphate Transporter (PHT) family protein in maize and analyzed the phylogenetic classifications, gene and protein architectures, CIS-regulatory element, and expression profile, among others. However, very similar work on the same crop has been published by earlier studies (https://doi.org/10.3390/ijms17060930). Therefore, I highly recommend the authors supplement their MS with more data. I recommend further analysis of the responsiveness of the PHT gene family on phosphorus uptake and partitioning under different conditions using qPCR techniques. Also, the amount of phosphorus taken up and accumulated in various tissues and genes expressed in those tissues should be determined. A working model can also be drawn to illustrate how the PTH gene regulates phosphorus uptake and homeostasis in maize.
Beyond the major concerns above, the MS has several issues that require authors' attention. There are many redundant phrases and duplications of sentences. I have pointed out some of these below for the author to consider in their revision.
Line 17: "in maize" is redundant, please delete
Lines 18-20: "In this study, we systematically identified the PHT gene family in maize, followed by phylogenetic, gene structure, and transcriptome analyses."
Revise as follows; "In this study, we systematically identified the PHT gene family in maize and analyzed their phylogenetic classification, gene structure, and expression profile."
Line 22: remove "of members"
Line 24: Whats "stage-"
Line 44: "In summary" This feels like a conclusion. Please start the sentence without it
Line 52: "phosphate transporter (PHT)" Once an abbreviation is defined at first use, the acronym should be used in the rest of the text.
Line 55 "the most extensively studied" please cite the most extensive sources
Line 60-61: "Studies have shown………." From the sentence, its not enough to have one citation here. Pls add the citation on the remaining studies of this gene.
Line 91-96: This sentence is too long and hard to read. Consider partitioning into 2 or 3
Lines 26 and 101: "This study represents the first comprehensive and systematic analysis of the ……" and "PHT1 gene family in maize has been studied …." These two sentences are contrasting
Obviously, this work gene family has been reported in maize, see https://doi.org/10.3390/ijms17060930
The claim of for as the first time may have to be changed
Line 111-114: split this sentence into 2
Line 115-116: Something is missing in this sentence
Line 123: replace "smaller" to "lower"
Lines 127-141: The entire paragraph describes protein secondary structures, but the manuscript does not include such an analysis
Line 154: arabidopsis capitalize "a"
Figure 3: Please align your phylogenetic tree to gene structures
Line 318 and 319: "The high expression ……….." consider placing these statements and put these sentences in proper context. There should be a background description of the expression levels of the genes. This has made the section appear more akin to discussion than results. Describes the expression data in a logical manner. The entire paragraph did not cite any Figure. Authors should show the specific expression results they are describing.
Line 336: The results……………………..response" Please restructure this sentence
Discussion. Author may consider adding the significance of the introns, exons, and subcellular localization of the PHT protein below the second paragraph. Please refer to the article below https://doi.org/10.1016/j.stress.2023.100214 and cite.
The authors need to clearly describe how their article differs from the previous genome-wide analysis article on maize.

Requires improvement
Author Response
Comment 1:
Title: Genome-Wide Identification and Expression Analysis of the Phosphate Transporter (PHT) Gene Family in Zea mays under Phosphorus Stress
Summary: The authors identified the Phosphate Transporter (PHT) family protein in maize and analyzed the phylogenetic classifications, gene and protein architectures, CIS-regulatory element, and expression profile, among others. However, very similar work on the same crop has been published by earlier studies (https://doi.org/10.3390 /ijms17060930). Therefore, I highly recommend the authors supplement their MS with more data. I recommend further analysis of the responsiveness of the PHT gene family on phosphorus uptake and partitioning under different conditions using qPCR techniques. Also, the amount of phosphorus taken up and accumulated in various tissues and genes expressed in those tissues should be determined. A working model can also be drawn to illustrate how the PTH gene regulates phosphorus uptake and homeostasis in maize.
Response 1:
Thank you for your valuable feedback and constructive suggestions. We appreciate the opportunity to clarify the novelty and scope of our study compared to earlier work. Below, we provide detailed responses to your comments:
- Regarding the suggestion to perform qPCR and phosphorus uptake experiments:
This is of course an interesting idea but beyond the scope of this study. This work is based on rigorously quality-controlled transcriptome data, which provide comprehensive insights into the PHT gene family under phosphorus stress and have effectively identified key genes. Hence, we believe the transcriptome analysis has already provided robust results and additional qPCR validation are likely to yield overlapping findings. Moreover, qPCR experiments targeting the responsiveness and partitioning as suggested by you, are a different objective and can be addressed in future research building on our findings presented in this work.
We employed whole-plant sampling to capture the plant's overall response to phosphorus stress, aligning with the genome-wide objectives of our study. While tissue-specific research was not included in the current scope, we fully acknowledge its importance. In future research, we plan to further explore tissue-specific sampling, combined with qPCR and physiological experiments, to validate the functions of key genes, investigate their expression patterns across different tissues, and elucidate their roles in phosphorus stress regulation.
Once again, we sincerely appreciate your suggestion, which has provided valuable direction for our future studies.
- Regarding the comparison with earlier studies (https://doi.org/10.3390/ijms170609 30):
We fully acknowledge the significance of previous studies on the maize PHT1 gene family and have appropriately cited them in our manuscript. However, our research differs from and expands upon these findings in several key aspects:
Broader Scope: Previous studies focused exclusively on the PHT1 subfamily and its role in arbuscular mycorrhizal (AM) symbiosis. In contrast, our study comprehensively analyzes all PHT subfamilies in maize (PHT1, PHT2, PHT3, PHT4, PHT5, and PHO1), providing an integrated perspective on the roles of this gene family in phosphorus stress response.
Phosphorus Stress Response: Unlike the cited research, our study emphasizes how the PHT gene family responds to phosphorus deficiency. We analyze their expression patterns under low-phosphorus conditions across different tissues and developmental stages.
Innovative Bioinformatics Approach: Our research incorporates comprehensive analyses of the transcriptomic data for all PHT genes, as well as protein-protein interaction networks, offering novel insights into the regulatory mechanisms of PHT genes under phosphorus stress.
Applications in Breeding: Our findings identify valuable molecular targets for improving phosphorus use efficiency in maize through breeding, an aspect that was not addressed in previous studies.
These major distinctions highlight the originality and importance of our research, which complements rather than duplicates earlier findings.
Comment 2:
Line 17: "in maize" is redundant, please delete
Response 2:
Thank you for pointing this out. We have removed the redundant phrase "in maize" from Line 17 to improve clarity.
Comment 3:
Lines 18-20: "In this study, we systematically identified the PHT gene family in maize, followed by phylogenetic, gene structure, and transcriptome analyses."Revise as follows; "In this study, we systematically identified the PHT gene family in maize and analyzed their phylogenetic classification, gene structure, and expression profile."
Response 3:
Thank you for your suggestion. We have revised the sentence as recommended by you.
Comment 4:
Line 22: remove "of members"
Response 4:
We have removed "of members" as requested to improve the sentence's clarity.
Comment 5:
Line 24: Whats "stage-"
Response 5:
Thank you for pointing this out. Following your comment, we have clarified the term by changing "stage-" to "developmental stage."
Comment 6:
Line 44: "In summary" This feels like a conclusion. Please start the sentence without it.
Response 6:
Thank you for your suggestion. We agree with your comment and have removed "In summary" to improve the flow and avoid the impression of a conclusion.
Comment 7:
Line 52: "phosphate transporter (PHT)" Once an abbreviation is defined at first use, the acronym should be used in the rest of the text.
Response 7:
Thank you for your suggestion. We have reviewed the manuscript and ensured that after the first use of "phosphate transporter (PHT)," the acronym "PHT" is consistently used throughout the text.
Comment 8:
Line 55 "the most extensively studied" please cite the most extensive sources.
Response 8:
Thank you for pointing this out. We have added the citation of Nussaume et al. (2011), which provides a comprehensive review of PHT1 transporters, summarizing their physiological, biochemical, molecular, and genetic roles in plant phosphate import.
Comment 9:
Line 60-61: "Studies have shown………." From the sentence, its not enough to have one citation here. Pls add the citation on the remaining studies of this gene.
Response 9:
Thank you for your suggestion. We have revised the sentence to "Research has shown………" to reflect that it refers to a single source rather than multiple studies.
Comment 10:
Line 91-96: This sentence is too long and hard to read. Consider partitioning into 2 or 3
Response 10:
Thank you for your suggestion. We have revised the sentence by breaking it into two shorter and more concise sentences for improved readability.
Comment 11:
Lines 26 and 101: "This study represents the first comprehensive and systematic analysis of the ……" and "PHT1 gene family in maize has been studied …." These two sentences are contrasting Obviously, this work gene family has been reported in maize, see https://doi.org/10.3390/ijms17060930 The claim of for as the first time may have to be changed.
Response 11:
Thank you for pointing this out. While there is no fundamental contradiction between these two statements, we acknowledge that there may be some ambiguity. To address this, we have revised the text to remove the word "first" for better clarity. We hope this revision resolves any potential confusion. Thank you again for your valuable feedback.
Comment 12:
Line 111-114: split this sentence into 2
Response 12: Thank you for your suggestion. We have revised the text by splitting the sentence into two for improved readability.
Comment 13:
Line 115-116: Something is missing in this sentence
Response 13: Thank you for pointing this out. We have revised the sentence to provide a more complete and accurate description. The updated text now reads: "Among the identified 34 PHT genes, the genes of the PHT1 subfamily were mainly located on chromosomes 1, 2, and 5, while other subfamilies exhibited a more scattered distribution across the maize genome." We hope this revision resolves the issue.
Comment 14:
Line 123: replace "smaller" to "lower"
Response 14:
Thank you for your suggestion. We have replaced "smaller" with "lower" as requested.
Comment 15:
Lines 127-141: The entire paragraph describes protein secondary structures, but the manuscript does not include such an analysis
Response 15:
Thank you for your observation. We focused on the predicted number of transmembrane helices, which were analyzed using the TMHMM-2.0 tool (https://services.healthtech.dtu.dk/services/TMHMM-2.0/). This information is provided in Table S1 and the analysis is detailed in the Materials and Methods section. The updated text in the revised manuscript explicitly states: "There are differences in the number of transmembrane helices among the ZmPHT subfamilies (Table S1). Transmembrane domain predictions were conducted using the TMHMM-2.0 tool.". We hope this clarification addresses your concern.
Comment 16:
Line 154: arabidopsis capitalize "a"
Response 16:
We have corrected "arabidopsis" to "Arabidopsis".
Comment 17:
Figure 3: Please align your phylogenetic tree to gene structures
Response 17:
Thank you for your suggestion. We have revised Figure 3 to align the phylogenetic tree with the gene structures for better clarity and visual consistency.
Comment 18:
Line 318 and 319: "The high expression ……….." consider placing these statements and put these sentences in proper context. There should be a background description of the expression levels of the genes. This has made the section appear more akin to discussion than results. Describes the expression data in a logical manner. The entire paragraph did not cite any Figure. Authors should show the specific expression results they are describing
Response 18:
Thank you for your suggestion. We have revised the sentences to place them in proper context and added a background description to clarify the expression levels of the genes. The revised text now reads: "The expression analysis revealed that ZmPHT3-4 and ZmPHO1-H3 showed notably high expression levels in root tissues compared to other genes in their respective subfamilies (Figure 6a). Their expression in root tissues suggests potential roles in phosphate uptake and transport." We hope this revision addresses your concerns by improving the clarity and logical flow of the results section.
Comment 19:
Line 336: The results……………………..response" Please restructure this sentence.
Response 19:
Response: Thank you for your suggestion. We have restructured the sentence to improve clarity and readability. The revised sentence now reads: "The results revealed showed a clear separation between the two lines under different phosphorus treatments, reflecting their contrasting phosphorus use efficiency. These differences highlighted the distinct transcriptional characteristics of the PHT gene family under low and moderate phosphorus conditions.” We hope this revision addresses your concern.
Comment 20:
Discussion. Author may consider adding the significance of the introns, exons, and subcellular localization of the PHT protein below the second paragraph. Please refer to the article below https://doi.org/10.1016/j.stress.2023.100214 and cite.
Response 20:
Thank you for your valuable suggestions! In the revised manuscript, we have added discussions on the significance of introns, exons, and subcellular localization of PHT proteins. Additionally, we have incorporated the suggested reference (https://doi.org/10.1016/j.stress.2023.100214) into the revised text. We believe these additions have enhanced the depth and relevance of the discussion. Once again, thank you for bringing this important point to our attention.
Comment 21:
The authors need to clearly describe how their article differs from the previous genome-wide analysis article on maize.
Response 21:
Thank you for your feedback! In the revised manuscript, we have added a section in the first part of the discussion. This section describes the differences between our study and the previous genome-wide analysis article in maize. We hope this addition highlights the novelty of our study and emphasizes its expanded scope compared to the previous research.

Reviewer 3 Report
Comments and Suggestions for Authors
The manuscript describes the genome-wide identification and expression analysis of the phosphate transporter (PHT) gene family in zea mays under phosphorus stress.
The text is well-written.
Some minor points are:
a) Ls. 124-6: Please, explain further.
b) L.234: Provide full names for ABRE, ARE, and explain “G-box elements”.
c) L.239: Provide full names for LTR and GC-motifs.
d) L.243: Provide full names forCGTCA and GARE.
e) L.486: Cite the previous study.
f) Please, include a comparative analysis between the findings in this work and those involving relative species, e.g., sorghum to point out similarities and/or differences.
Author Response
Comment 1:
Ls. 124-6: Please, explain further.
Response 1:
Thank you for your suggestion. We have revised and expanded this section to provide a more detailed explanation.
Comment 2:
L.234: Provide full names for ABRE, ARE, and explain “G-box elements”
Response 2:
Thank you for your suggestion. We have revised the text to include the full names of ABRE and ARE, and provided a detailed explanation of the G-box elements. Specifically:
ABRE: abscisic acid responsive element
ARE: anaerobic responsive element
G-box elements: a cis-regulatory element involved in light-responsive transcriptional regulation
These additions can be found in lines 249-250 in the revised manuscript. We hope these clarifications improve the clarity and understanding of the article.
Comment 3:
L.239: Provide full names for LTR and GC-motifs.
Response 3:
Thank you for your suggestion. We have revised the text to include the full names and functional descriptions of LTR and GC-motifs. Specifically:
LTR: "Low Temperature Response Element," a cis-regulatory element involved in regulating gene expression under low-temperature conditions.
GC-motifs: an enhancer-like cis-regulatory element involved in specific gene induction under anoxic conditions.
These additions can be found on Lines 255-256 in the revised manuscript. We sincerely appreciate your valuable suggestion!
Comment 4:
L.243: Provide full names forCGTCA and GARE.
Response4:
Thank you for your suggestion. We have revised the text to include the full names and functional descriptions of CGTCA-motif and GARE-motif. Specifically:
CGTCA-motif: a cis-acting regulatory element involved in methyl jasmonate (MeJA) responsiveness.
GARE-motif: a cis-regulatory element mediating transcriptional regulation in response to gibberellin signaling.
These additions can be found on Lines 260-262 in the revised manuscript. We sincerely appreciate your valuable suggestion!
Comment 5:
L.486: Cite the previous study.
Response 5:
Thank you for your suggestion. We have added the relevant citation in Line 486 to support the rationale for the parameter settings used in the BLASTP analysis. The full citation has been included in the reference list (see reference number [60]).We appreciate your valuable input!
Comment 6:
Please, include a comparative analysis between the findings in this work and those involving relative species, e.g., sorghum to point out similarities and/or differences.
Response 6:
Thank you for your suggestion. We have conducted a comparative analysis between our findings on maize PHT gene family members and the study on sorghum by Wang et al. (2019), and the revised content has been included in Section 3.4 of the discussion. The reference has also been added to the bibliography.
